# Neglected Monteggia Fractures in Children—A Retrospective Study

**DOI:** 10.3390/children9081100

**Published:** 2022-07-22

**Authors:** Dragoljub Zivanovic, Zoran Marjanovic, Nikola Bojovic, Ivona Djordjevic, Maja Zecevic, Ivana Budic

**Affiliations:** 1Faculty of Medicine, University of Nis, 18000 Nis, Serbia; zoran.marjanovic@medfak.ni.ac.rs (Z.M.); ivona.djordjevic@medfak.ni.ac.rs (I.D.); ivana.budic@medfak.ni.ac.rs (I.B.); 2Clinic for Pediatric Surgery, Pediatric Orthopedics and Traumatology, University Clinical Centre Nis, 18000 Nis, Serbia; niboj.nb@gmail.com (N.B.); maja.zecevicmd@gmail.com (M.Z.); 3Clinic for Anesthesia and Intensive Therapy, University Clinical Centre Nis, 18000 Nis, Serbia

**Keywords:** Monteggia fracture, children, neglected, chronic, ulnar osteotomy, annular ligament

## Abstract

(1) Background: A Monteggia fracture is an infrequent injury in children. It can be missed during an initial consultation in 20–50% of patients. Chronic radial head dislocation may lead to several complications. Thus, surgical reconstruction of chronic Monteggia injuries is justified. The aim of this study is to analyze the single tertiary center experience in the treatment of neglected Monteggia fractures. (2) Methods: A retrospective study of patients treated for missed Monteggia fractures was conducted. Hospital records, operative charts, follow-up records and a set of X-rays were analyzed for each patient. Radiographic results were graded as good, moderate or poor. The functional status of elbows was estimated using the Mayo Elbow Performance Index. (3) Results: A total of 13 patients (8 boys and 5 girls) aged 4–12 years (mean 7.15) were treated during the study period. An angulation osteotomy of the ulna was performed in ten patients and a radial shortening osteotomy in three patients. A Bell–Tawse annular ligament reconstruction was performed in five patients, and a direct repair was performed in two patients. Eight patients had radiocapitellar trans-fixation. There were nine good radiographic results, three moderate and one poor. The functional result was excellent in nine patients, good in three and poor in one. (4) Conclusions: Our work has many limitations (only 13 patients and different types of operations), and conclusions should be drawn very carefully from such a small and diverse group. The surgical reconstruction of neglected Monteggia fractures in children should be attempted in all patients. Angulation and elongation osteotomies of the ulna are suitable for most patients. If there is a marked overgrowth of the radius, gradual ulnar lengthening and radial head reduction using the Ilizarov method may be a better option. Annular ligament reconstruction is not mandatory.

## 1. Introduction

The specific injury consisting of a proximal ulna fracture accompanied by dislocation of the radial head, which was first described and published by Giovanni Battista Monteggia in 1814, was, in fact, a description of a neglected injury. The majority of Monteggia fractures in children, if properly diagnosed at the time of injury, may be treated conservatively with closed reduction and cast immobilization. However, in cases of irreducible radial head dislocation or unstable fracture of the ulna, surgery may be necessary [1]. The most serious complication of this injury is the failure to recognize it. Radial head dislocation is often missed [2], and injury is treated as a simple fracture of the ulna, an extremely rare injury in childhood. Moreover, plastic deformation of the ulna may be left unrecognized, and the injury may be treated as an isolated radial head dislocation [3], which is another extremely rare condition in children. Strengthening the physiological ulnar curve in these patients may prevent accurate reduction of the radial head or result in incomplete reduction [2,3,4,5]. Furthermore, re-dislocation of the reduced radial head or late dislocation in patients with initially normal radiographs has been described in the literature [6]. Permanent dislocation of the radial head, which is considered chronic after four weeks [7], can lead to several further complications such as palpable mass, a decrease in elbow flexion and forearm pronation and supination, valgus elbow deformity, instability of the elbow joint and late (tardy) ulnar palsy [8,9,10,11]. While treating acute Monteggia fractures in children is mostly straightforward, the treatment of neglected chronic lesions is controversial. Several studies have demonstrated that the natural course of chronic radial head dislocation is not favorable and that conservative treatment is not acceptable [10,12]. A variety of surgical options for the correction of neglected Monteggia lesions in immature patients have been described: open reduction of the radial head with or without annular ligament reconstruction, various ulnar or radial osteotomies or gradual reduction of the radial head by elongation of the ulna with an external fixator [10,12,13,14].

This study aims to analyze the single-center experience of treating missed Monteggia fractures in childhood.

## 2. Materials and Methods

We performed a retrospective analysis of the medical records of pediatric patients treated for neglected Monteggia fractures between January 2000 and December 2020 in a tertiary university children’s hospital. An institutional ethics committee approved this retrospective study (approval protocol code 11063, date of approval 19 April 2022). Hospital records, operative charts, follow-up records and sets of X-rays for each patient were analyzed. Data were collected and analyzed in compliance with the Helsinki Declaration.

The inclusion criteria were as follows: chronic Monteggia injury in patients of both sexes and younger than 18 years of age. We considered Monteggia injury as chronic 4 weeks post-injury, similarly to most researchers, although there is no consensus in the literature about this issue.

The exclusion criteria were acute Monteggia fractures with an interval shorter than 4 weeks from the injury, Monteggia equivalent fractures, patients with congenital radial head dislocation, patients with incomplete medical documentation and cases with a follow-up period less than 12 months.

A retrospective search of the medical records identified 13 children with missed Monteggia fractures that met the inclusion criteria for the study period. None met the exclusion criteria, and all 13 patients were included in the study.

There were 8 boys (62%) and 5 girls (38%), with the right side affecting 4 children and the left in 9 children. The mean age of the patients at the time of injury was 7.15 years (range 4–12 years).

Osteotomy

An ulnar osteotomy was performed in ten children. We used a straight incision directed over the planned osteotomy site. Eight osteotomies were performed high in the metaphyseal region of the ulna, and the remaining two were performed distally at the level of maximal angulation of the ulna. In all cases, ulnar osteotomies were stabilized with a plate and at least four screws.

A radial shortening osteotomy was performed in three patients with long-lasting radial head dislocation and overgrowth of the radius. Fixation of the radial osteotomy was accomplished with a plate and screws in two children and an intramedullary Kirschner wire in one child.

2.Open reduction of the radial head

In three patients, stabile reduction of the radial head was achieved by closed means after the ulnar osteotomy alone. If a dislocated radial head could not be reduced or reduction could not be maintained after an osteotomy of the ulna, we proceeded to open reduction via a separate curved J incision. In 10 patients, open reduction of the radiocapitellar joint with debridement of interposed soft tissue was necessary; in another 7 patients, it was necessary after osteotomy of the ulna and in a further 3 patients was necessary after osteotomy of the radius.

3.Annular ligament repair

Repair of the annular ligament was made at the surgeons’ discretion. In five patients, the annular ligament was repaired with the lateral strip of triceps fascia after open reduction of the radial head, according to the Bell–Tawse technique. Direct suture of the torn annular ligament was performed in 2 patients, and in 2 cases, repair of the annular ligament was not attempted after open reduction of the radial head.

4.Transcapitellar fixation

Stability of the radial head after open reduction was secured with transcapitellar heavy Kirschner wire in 8 patients: 3 after a radial shortening osteotomy and 5 after an ulnar angulation osteotomy. In 7 of these patients, repair or reconstruction of the annular ligament was also performed.

5.Postoperative immobilization

In all 13 patients, a long-arm cast was applied with the elbow in 90° of flexion and the forearm in a neutral rotation or slight pronation of up to 10°. Casts were removed after 4–6 weeks. Transcapitellar Kirschner wire, if applied, was removed at the same time under local anesthesia. Patients were then allowed free elbow motion as tolerated. All 13 patients underwent intensive physical therapy. Physical and sports activities were restricted for 3 months after cast removal. Plates and screws were removed 6–12 months postoperatively.

6.Follow-up

Patients were followed up at three-month intervals in the first year after surgery and then yearly. Reduction and congruity of the radiocapitellar joint and healing of osteotomies were analyzed on AP and lateral radiographs. Radiographs were obtained postoperatively at the time of cast removal and on follow-up visits at 3, 6 and 12 months. Radiographic results were graded as good (complete reduction of radial head and no arthritic changes), moderate (persistent subluxation and/or arthritic changes) and poor (complete dislocation of the radial head) as described by Delpont et al. [14].

7.Functional assessment

On follow-up visits, elbow stability, range of elbow flexion and extension, forearm pronation, supination, and carrying angle were measured and compared to the uninjured side. Patients and their parents were asked about pain, discomfort and limitations in everyday life activities or difficulties in sports activities. Functional results were assessed using the Mayo Elbow Performance Index (MEPI) [15].

## 3. Results

Thirteen patients with neglected Monteggia fractures were identified in the twenty-year period. There were eight boys and five girls. The mean age of the patients was 7.15 years (range 4–12 years). The right side was injured in four children and the left in nine children. None of the patients had a history of previous injuries to affected elbows and forearms. Two children had no treatment for the initial injury. Their parents did not consider the initial injuries serious enough to request medical help. Nevertheless, they presented 14 and 36 months after the injury with a visible and palpable mass in the anterior elbow region, pain, limitation of elbow flexion and forearm rotation and valgus deformity of the elbow. Eleven patients were initially treated with an above-elbow cast because their injuries were misdiagnosed as isolated ulna fractures. In six of them, radial head dislocation was observed after cast removal. Besides the children having a palpable mass, pain, a limited range of motion and dislocated radial head on X-rays, two children with lateral radial head dislocation (Bado type III) had severe paresis of the motor branch of the radial nerve. In another four patients, chronic radial head dislocation was noticed after physical therapy, when no improvements in the range of motion were observed. Dislocation of the radial head was obvious in initial postinjury X-ray studies in all these patients. Finally, one patient had his Monteggia fracture initially treated with closed reduction of the displaced radial head and closed reduction and elastic stabile intramedullary fixation (ESIN) of the ulnar fracture with a titanium elastic nail (TEN). Secondary displacement of the radial head, due to insufficient fixation of the ulna, was left unrecognized until cast removal four weeks later. A summary of patients’ demographics and treatment overview is given in Table 1.

The average time from injury to surgical treatment of the neglected Monteggia fracture was 7.6 months (range 1–36 months).

Nine patients had a Bado type I injury, with the radial head displaced anteriorly, and four patients had a Bado type III injury, with lateral and anterior radial head displacement. All patients were treated as soon as possible after a diagnosis of the missed Monteggia fracture was established. The period from diagnosis to surgery varied from 2 days to 2 weeks. An autologous bone graft, harvested from the iliac crest, was used in only one case to fill the gap after osteotomy of the ulna. Another nine osteotomies of the ulna and three osteotomies of the radius healed without bone grafting. All osteotomies of the ulna and two osteotomies of the radius were stabilized with plates and 4–8 screws (Figure 1 and Figure 2).

One radial osteotomy was fixed with intramedullary Kirschner wire, which was simultaneously used for the transcapitellar fixation. No cases of hardware failure were observed. Moreover, there were no cases of breakage of the transcapitellar Kirschner wires. There were no cases of delayed union or nonunion. All osteotomies of the ulna and radius were healed on radiographic controls 3 months after surgery. In two patients with long-lasting radial head dislocation, heterotopic ossification had developed prior to surgery. Ossifications were excised during debridement of the radiocapitellar joint and open reduction of the radial head. We did not observe a difference in radiocapitellar joint stability between patients with or without reconstruction of annular ligaments.

Complete re-dislocation of the reduced radial head occurred in two patients. In the first patient, dislocation occurred within 2 weeks after cast and transcapitellar Kirschner wire removal. Revision surgery was performed. The displaced radial head was reduced again with a shortening osteotomy of the radius and transcapitellar Kirschner wire fixation. Unfortunately, radial head dislocation re-occurred after cast and Kirschner wire removal six weeks later. The child underwent two more surgeries in another hospital but with poor results. In the second patient, re-dislocation of the radial head occurred 2 days after the ulnar osteotomy and closed reduction of the radial head. Open reduction of the dislocated radial head, debridement of the radiohumeral joint and direct repair of the annular ligament, without revision of the ulnar osteotomy, were performed. Three patients had a residual radial head subluxation but without significant functional impairment. The radiographic and clinical results are summarized in Table 2.

At the one-year follow-up visit, the mean elbow extension was −4.00° (range 0° to −17°). Extension was limited in four patients, but the limitation was mild in two of them. The average elbow flexion was 127.31° (range 140°–90°). Serious loss of flexion occurred in one patient (case no. 3) with a poor overall result. All patients encountered some degree of reduction in their range of pro-supination, which was usually mild (Figure 3). Pronation, with an average of 77.92° (range 90°–27°), was less affected than supination, which had a mean of 75.46° (range 90°–10°).

The average MEPI was 92.31 (range 50–100). The lowest score, as was expected, was in patient no. 3, who had recurrent radial head dislocations. According to MEPI, nine patients had excellent functional results, three good and one poor. In two patients with good results, the MEPI score was 85 because both children had complained about mild pain after strenuous physical activities. The third patient with a good functional result had mild pain and an arc of motion between 50° and 100°.

## 4. Discussion

Monteggia fractures are uncommon injuries in children. Failure to recognize a Monteggia fracture on initial consultation occurs in 20–50% of cases [6]. Radial head dislocation is usually overlooked either because of focus on the ulnar fracture or an incorrect set of radiographs on which the elbow joint has not been clearly visualized. Persistent dislocation of the radial head results in progressive dysplastic changes of the radiocapitellar joint due to a lack of joint restraint [8]. In long-lasting dislocations, the radial head loses its concave articular surface and displays hypertrophic changes, and the humeral capitellum flattens, limiting the range of elbow flexion and extension [12]. Restricted forearm pronation and supination, pain, palpable mass, valgus deformity of the elbow and neurological complications may occur as a consequence of untreated radial head dislocation [9,16,17]. All these symptoms were present in our patients. Therefore, we agree with De Gennaro et al. that there is no space for a wait-and-see policy in current practice [9]. Another source of neglect of the Monteggia injury is the failure to recognize plastic deformation of the ulna in children with apparent radial head dislocation (so-called Monteggia variant) [6]. In fact, the majority of the cases previously considered congenital radial head dislocations were posttraumatic in origin with evidence of plastic deformation of the ulna; so-called “traumatic ulnar bowing” may be demonstrated in almost all cases [3,5]. Even birth trauma may lead to traumatic ulnar bowing with consequential radial head dislocation, which could easily be erroneously considered to be congenital in origin. True congenital radial head dislocations are usually bilateral, with a tendency of familial involvement and are commonly associated with dysplasia of the radial head and capitulum [18]. We did not observe patients with chronic radial head dislocation due to neglected plastic deformation of the ulna.

There is still no consensus on the management of neglected Monteggia fractures. Some authors recommend surgical procedures directed to the radiocapitellar joint [8,19]. The radial head may be irreducible after 3 weeks due to fibrosis [14], and open reduction with joint debridement and excision of annular ligament remnants may be necessary. Authors who have taken the opposite approach focus on ulnar osteotomy with correction or overcorrection of deformities of the ulna [2,8,10,15,20]. The main disadvantage of this procedure is unfavorable cosmetic appearance if overcorrection of the ulna has to be performed [21]. Several authors have recently reported excellent results in relation to treatment with gradual ulnar lengthening and radial head reduction with external fixators [13,21,22]. Chen et al. reported that a radial shaft shortening osteotomy was performed in two cases to reduce tension and keep the radial head in a stable reduction in addition to the ulnar osteotomy [23]. Deformity of the ulna in 10 of our patients required correction by osteotomy. In three other cases, open reduction of the radial head was performed. In all three patients, a shortening radial osteotomy was performed because of radial overgrowth due to long-lasting dislocation of the radial head. In one of them (case no. 2), there was persistent radial head subluxation. Despite only a moderate radiographic result and some limitations in elbow and forearm movements, the patient’s functional impairment is minor. He is now an adult, a physical worker and complains of mild pain after strenuous activities. Nevertheless, for him and the patient in whom re-dislocation of the radial head occurred (case no. 3), gradual lengthening of the ulna and radial head reduction with an external fixator may represent a reasonable alternative treatment option. We do not have personal experience with this method, but the published results are encouraging. We assume that gradual lengthening and angulation of the ulna, with a reduction of radial head dislocation with an external fixator, should always be considered as a treatment option in patients with long lasting dislocation and radial overgrowth.

All the patients in our study were operated on shortly after being diagnosed with a chronic Monteggia lesion. However, there is no need to hurry. Thorough preoperative planning of the reconstructive procedure is very important. A set of correct radiographs in true AP and lateral projection of both injured and uninjured forearms should be obtained [9]. If any discrepancy in the physiological curve of the ulna exists, it should be corrected by ulnar osteotomy. Straightening the natural ulnar curve may result in an incomplete reduction of the radial head, re-dislocation or even late dislocation [24]. We observed re-dislocation of a reduced radial head in one patient as the cause of their chronic Monteggia lesion. His ulna fracture was originally stabilized with a retrograde elastic titanium nail, but progressive loss of reduction with angulation of the ulna occurred. That was left unnoticed until the follow-up for cast removal, which was 4 weeks after the surgery. Revision and fixation with a plate and screws were subsequently performed. We believe that retrograde ESIN may provide insufficient stabilization of the ulnar fracture in a Monteggia injury. Anterograde ESIN through the olecranon entry point may be a better solution. Alternatively, open reduction and plating of the ulna may be performed. If malalignment of the ulna exists, then an osteotomy of the ulna should be the first step in the treatment of chronic Monteggia fractures, especially 3–4 months from the initial injury. A posterior angulation and elongation osteotomy of the ulna, as described by Bouyala [25], is currently the preferred method. Performing an osteotomy high in the metaphyseal region of the proximal third of the ulna may decrease the risk of delayed union or nonunion. Eight of our patients were treated this way (Figure 1). Other authors prefer a more distal osteotomy at the site of maximal angulation of the ulna [26]. Sliding [2] and “Z”-lengthening [27] osteotomies of the ulna have been recently published. In two of our patients, correction of ulnar angulation was performed at the level of a fracture of the ulna. Both patients underwent surgery 5 weeks after their injury, and their fractures, although consolidated, were not completely healed; we decided to correct the deformity at that level (Figure 2). Kim et al. [28] recently performed a 3-dimensional analysis of the deformation of the ulna in patients with chronic radial head dislocation. They found not only angulation but a complex torsional deformity of the ulna. This may explain why a simple angulation-lengthening osteotomy of the ulna may not be completely effective in all cases, especially in Bado type III cases, as reported by Delpont et al. [14] and was observed in one of our cases. Correction of such a complex deformity requires complex correction in two or three planes.

Another source of controversy is the value of reconstruction of the annular ligament. Some authors, such as Nakamura [12] and David-West [6], recommend it in all cases. Others, such as Devnani and Laderman [8,29], negate its usefulness entirely. Mild restriction in forearm pro-supination was observed in most of our patients who underwent annular ligament reconstruction. Two patients without ligament reconstruction did not experience complications, and re-dislocation of the radial head occurred in one patient. If stabile closed reduction of the radial head could be achieved by ulnar osteotomy alone, we would no longer insist on the annular ligament repair, as Bhaskar [30] previously reported. If the radial head is unstable, open reduction and debridement of the radiocapitellar joint and repair or reconstruction of the annular ligament are recommended. If direct repair is not possible, our preferred method would be a Bell–Tawse reconstruction with a stripe of triceps fascia, although Garg [31] reported better results with the use of a palmaris longus graft. Finally, in patients with a reconstruction of the annular ligament, transcapitellar fixation with heavy Kirschner wire may provide additional stability and decrease the risk of radial head re-dislocation [6,9,32], despite Delpont [14] considering it to be contraindicated. In our study, radiocapitellar fixation was performed in all patients with annular ligament repair or reconstruction.

For the clinical evaluation of elbow function, we used the Mayo Elbow Performance Index (MEPI), although Kim’s elbow performance score, the Oxford elbow score and modified DASH could be used for the same purpose [9,10,33]. The MEPI results of our patients are comparable to reports of Nakamura, Delpont and Datta [12,14,15], who used the same score for the assessment of elbow function in their studies.

Our study has many limitations. First, as with all retrospective studies, it suffers from data inconsistency. Second, the number of patients is small, only 13 patients, to draw strong conclusions, but most published studies have a similar disadvantage. Only two published studies have reported larger groups of patients [31,34]. Third, there was a lack of data on the preoperative assessment of elbow function for some patients, so we were unable to compare the improvement in elbow function with operative correction. Fourth, patients had very different types of operations. Finally, only two of our patients reached skeletal maturity at the last follow-up. A well-designed, prospective multicenter study in the future is probably the best way to provide a sufficient number of patients and enable researchers to draw strong conclusions and treatment recommendations for this rare injury.

## 5. Conclusions

Our work has many limitations, and conclusions should be drawn very carefully from such a small and diverse group.

The best option for a neglected Monteggia injury is prevention. For patients presenting with neglected Monteggia fractures, reconstruction should be attempted in all cases. Angulation and elongation osteotomies of the ulna at the proximal third should be considered as the first option. If stabile reduction of the radial head is achieved after the ulnar osteotomy, reconstruction of the annular ligament and transcapitellar fixation are not recommended. If reduction of the radial head is not possible or is unstable, open reduction and joint debridement is indicated. In cases where completely stabile reduction of the radial head could not be achieved after open reduction and an ulnar osteotomy, direct repair of the annular ligament or reconstruction, preferably using a Bell–Tawse procedure, may be considered. Transcapitellar fixation with heavy Kirschner wire may provide additional stability in such cases. For patients with long-lasting dislocation and overgrowth of the radius, gradual reduction of the radial head and lengthening of the ulna with an external fixator (unilateral or circular) should be considered rather than a shortening osteotomy of the radius.

## Figures and Tables

**Figure 1 children-09-01100-f001:**
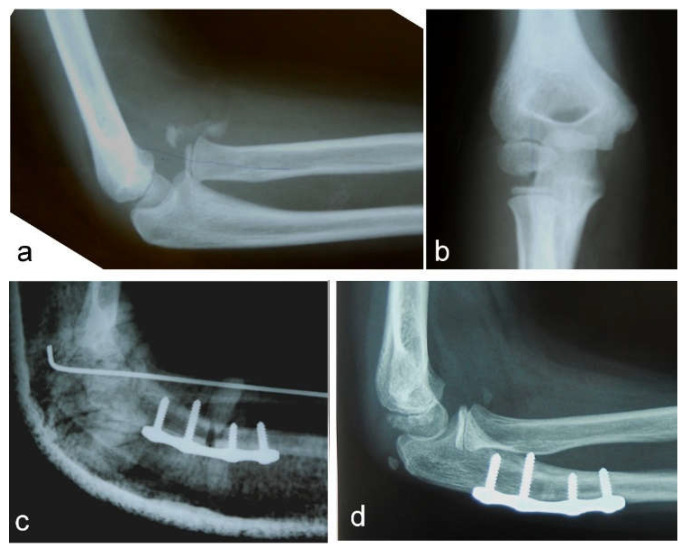
Neglected Monteggia fracture: (**a**) anterior dislocation of the radial head with heterotopic ossification visible on a lateral view; (**b**) AP view; (**c**) high posterior angulation and elongation osteotomy of the ulna stabilized with a plate and four screws. Transcapitellar fixation of the radial head with Kirschner wire; (**d**) the reduced radial head and healed ulnar osteotomy are visible on a lateral X-ray before removal of the plate and screws.

**Figure 2 children-09-01100-f002:**
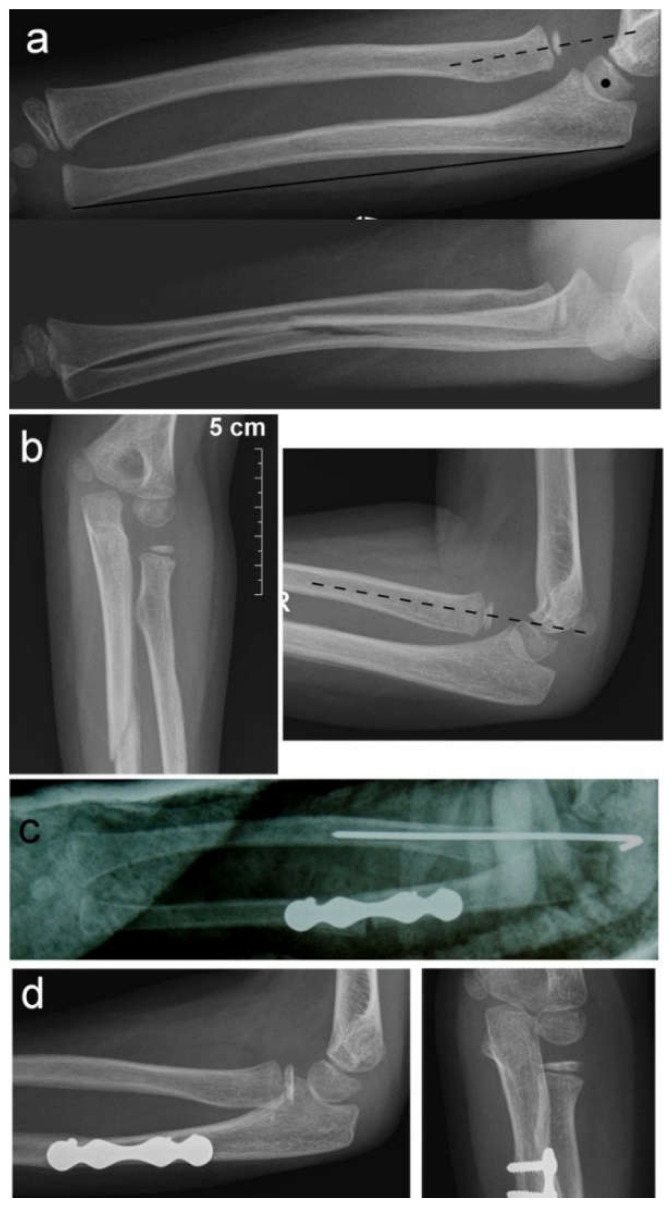
Ulnar osteotomy for chronic Monteggia lesion. (**a**) Radial head dislocation (dotted line) and ulnar angulation (heavy line) left unnoticed on initial X-rays; (**b**) chronic radial head dislocation and consolidation of ulnar fracture after cast removal at 4 weeks; (**c**) angulation osteotomy at the level of the fracture stabilized with a plate and screws and transcapitellar radial head fixation; (**d**) healed ulnar osteotomy and reduced radial head prior to hardware removal.

**Figure 3 children-09-01100-f003:**
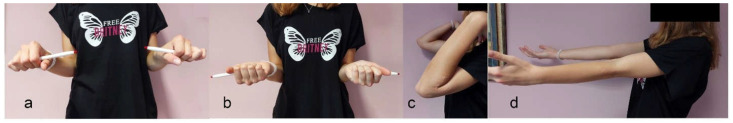
Thirteen years after reconstruction of the chronic Monteggia fracture. (**a**) Forearm pronation; (**b**) forearm supination; (**c**) elbow flexion and (**d**) elbow extension.

**Table 1 children-09-01100-t001:** Patients’ demographics and treatment overview.

Case No.	Sex	Side	Bado Type	Age at Surgery (Years)	Time from Injury to Diagnosis (Months)	Treatment			
						Reduction of Radial Head	Transcapitellar Kirschner Wire Fixation	Annular Ligament Repair or Reconstruction	Osteotomy
1	M	R	I	7	6	Open	Yes	Bell-Tawse	Radial shortening
2	M	L	I	12	36	Open	Yes	Bell-Tawse	Radial shortening
3	M	L	I	9	14	Open	Yes	Bell-Tawse	Radial shortening
4	M	L	I	4	1	Open	No	No	Ulnar angulation
5	F	R	I	6	1	Open	Yes	Direct repair	Ulnar angulation
6	M	L	III	10	5	Open	No	No	Ulnar angulation
7	F	L	I	7	2	Open	Yes	Bell-Tawse	Ulnar angulation
8	F	R	I	5	3	Open	Yes	Bell-Tawse	Ulnar angulation
9	M	R	I	10	5	Closed	No	No	Ulnar angulation
10	M	S	I	5	1.5	Closed	No	No	Ulnar angulation
11	F	S	III	6	2	Open	Yes	Direct repair	Ulnar angulation
12	M	S	III	4	1.5	Open	Yes	No	Ulnar angulation
13	F	S	III	8	1	Closed	No	No	Ulnar angulation

**Table 2 children-09-01100-t002:** Radiographic and clinical results.

Case No	Radiographic Result	Elbow Flexion/ExTension	Forearm Pronation/Supination	MEPI *	
				Score	Result
1	Good	135/0	80/85	100	Excellent
2	Moderate	124/0	75/70	85	Good
3	Poor	90/−17	27/10	50	Poor
4	Good	127/0	83/78	100	Excellent
5	Moderate	130/−10	71/64	100	Excellent
6	Good	140/0	82/79	85	Good
7	Good	134/0	90/85	100	Excellent
8	Good	130/0	90/80	100	Excellent
9	Good	130/−5	85/80	100	Excellent
10	Good	140/0	90/85	100	Excellent
11	Moderate	140/0	85/85	100	Excellent
12	Good	135/−5	85/90	100	Excellent
13	Good	100/−15	70/90	80	Good

* MEPI Mayo Elbow Performance Index.

## Data Availability

The data are available in the results section of the manuscript.

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
