# Peer review of "Neglected Monteggia Fractures in Children—A Retrospective Study"

_children, 2022, doi:10.3390/children9081100_

Round 1
Reviewer 1 Report
This study described clinical results of neglected Monteggia lesions, which are rather uncommon but still often seen in clinical setting. It may be difficult to describe a large scale results for neglected Monteggia lesions. Under these conditions, this study collected altogether 13 patients with neglected Monteggia lesion. I appreciate the authors’ efforts to complete this paper.
The point I have to mention is that the forearm pronation/supination must be measured at the elbow in flexed position. If it is measured at the elbow in extended position, rotational joint movement of the shoulder can compensate or simulate forearm rotation. Therefore, Fig. 3 and 4 are inappropriate.
Author Response
Dear reviewer,
We appreciate very much for your valuable comments and suggestions.
Forearm pronation and supination were measured with elbow in 900 of flexion in our study. I agree that figures 3 and 4 might be inappropriate because forearm pronation and supination is presented with elbows in extension, although in several published reports on the same topic pronation and supination were also presented with elbow in full extension on clinical photographs. Nevertheless, figure 3 is replaced with new figure where forearm pronation and supination are presented with elbow in 900 of flexion. Figure 4 has been deleted.
We are ready to address any further suggestions.
Reviewer 2 Report
The authors did not show anything new in the manuscript. There are many articles available on related topics. A small research group - only 13 people, a very large age range (4-12 years), comparing various surgical techniques: Angulation osteotomy of ulna 20 was performed in 10 and radial shortening osteotomy in 3 patients. Bell-Tawse annular ligament 21 reconstruction was performed in 5 and direct repair in 2 patients. Eight patients had radiocapitellar 22 trans-fixation. A very diverse research group is a mistake. The group is too small and diverse to draw conclusions. Nothing new for orthopedists is presented.Author Response
Dear reviewer,
We appreciate very much for your valuable comments and suggestions.
Neglected Monteggia injury is rare, yet important condition and collecting enough patients to draw strong conclusions may be difficult. To the best of our knowledge only two studies with more than 50 pediatric patients were published so far. Most of the published studies reports 10-20 patients. In our opinion, only well designed multicenter prospective study may improve limitations of our and many other studies with small number of participants and diverse methods of treatment and help researchers to draw strong conclusions and treatment recommendations. Nevertheless, we consider that our study as well as other studies with small number of patients and diverse treatment modalities may add very small but still useful piece of data to overall community. We try to improve our manuscript according to your valuable comments.
We are ready to address any further suggestions.
Corresponding author,
Reviewer 3 Report
Was there institutional ethics approval for this retrospective chart review? If so, this should be stated in the methods section.
In the methods section, the authors should state how many patients were eligible for inclusion in the study and how many were eventually included. This should be supported by a clear statement of the inclusion and exclusion criteria used for this study.
Author Response
Dear reviewer,
Authors appreciate very much your valuable comments and suggestions.
Institutional Ethics Committee have approved our retrospective study. Statement of ethics committee approval was entered separately through submission process according to Journal requirements. We added statement of ethics committee approval in Methods sections as you suggested in the review.
In the Methods section we added number of eligible patients and number of patients included in the study. All 13 eligible patients were included.
A clear statement of inclusion and exclusion criteria was also added to Methods section. We admit that inclusion and exclusion criteria were not clearly stated in original version of the manuscript.
We are ready to address any further suggestions.
Corresponding author
Round 2
Reviewer 2 Report
The authors, in the limitations of the work in the conclusions and in the abstract, must clearly emphasize that: the work has a lot of limitations - only on 13 patients and that the patients had very different types of operations. Therefore, conclusions should be drawn very carefully from such a small and so diverse group.Author Response
Dear reviewer,
In accordance with your comments and suggestions authors have corrected submitted manuscript.
We added clear statement that our work has a lot of limitations (only 13 patients and different types of operations used) and that conclusions should be drawn very carefully from such a small and so diverse group in abstract, conclusion and in discussion where limitations of our study were explained.